

# EOT20: A global ocean tide model from multi-mission satellite altimetry

Michael G. Hart-Davis[1], Gaia Piccioni[1,2], Denise Dettmering[1], Christian Schwatke[1], Marcello Passaro[1], and Florian Seitz[1]

[1]Deutsches Geodätisches Forschungsinstitut der Technischen Universität München (DGFI-TUM), Arcisstrasse 21, 80333 München, Germany
[2]Now at: Enel Global Trading S.p.A. Address: V.le Regina Margherita 125, 00198 Rome, Italy

**Correspondence:** Michael G. Hart-Davis (michael.hart-davis@tum.de)

**Abstract.**

EOT20 is the latest in a series of empirical ocean tide (EOT) models derived using residual tidal analysis of multi-mission satellite altimetry at DGFI-TUM. The amplitudes and phases of seventeen tidal constituents are provided on a global 0.125-degree grid based on empirical analysis of seven satellite altimetry missions and four extended missions. The EOT20 model

shows significant improvements compared to the previous iteration of the global model (EOT11a) throughout the ocean, particularly in the coastal and shelf regions, due to the inclusion of more recent satellite altimetry data as well as more missions, the use of the updated FES2014 tidal model as a reference to estimated residual signals, the inclusion of the ALES retracker and improved coastal representation. In the validation of EOT20 using tide gauges and ocean bottom pressure data, these improvements in the model compared to EOT11a are highlighted with the root-square sum (RSS) of the eight major tidal constituents

improving by $\sim 3$ cm for the entire global ocean with the major improvement in RSS ($\sim 3.5$ cm) occurring in the coastal region. Concerning the other global ocean tidal models, EOT20 shows an improvement of $\sim 0.2$ cm in RSS compared to the closest model (FES2014) in the global ocean. Variance reduction analysis was conducted comparing the results of EOT20 with FES2014 and EOT11a using the Jason-2, Jason-3 and SARAL satellite altimetry missions. From this analysis, EOT20 showed a variance reduction for all three satellite altimetry missions with the biggest improvement in variance occurring in the coastal

region. These significant improvements, particularly in the coastal region, provides encouragement for the use of the EOT20 model as a tidal correction for satellite altimetry in sea-level research. All ocean and load tide data from the model can be freely accessed at https://doi.org/10.17882/79489 (Hart-Davis et al., 2021).

## 1 Introduction

The regular fluctuations of the sea surface caused by ocean tides have intrigued and fascinated scientists for centuries based

on its influence on oceanic processes. Understanding ocean tides is vital for a variety of geophysical fields, with it being of particular importance in studies of the coastal environment and ocean mixing. Precise knowledge of ocean tides is also important for satellite altimetry and in determining high-resolution temporal gravity fields from, for example, the GRACE missions (Tapley et al., 2004).



In certain studies of non-tidal signals using satellite altimetry data, such as in sea-level and ocean circulation research, ocean
tides need to be removed from the data signal to properly study these processes. These so-called tidal corrections are usually
provided by ocean tide models that have been specially developed to predict the tidal signals throughout the global ocean.
The ever-evolving and improving field of ocean tide modelling has resulted in significant leaps in the accuracy of estimations
of ocean tides (Shum et al., 1997; Stammer et al., 2014). There are several ocean tide models that have been developed using
different techniques and for different applications, with a comprehensive summary of these models being presented in Stammer
et al. (2014). In general, ocean tides are known in the open ocean region to an accuracy of approximately 2 cm (Savcenko and
Bosch, 2012), however, models show large discrepancies between one another and compared to in-situ observations in the
coastal region (Ray et al., 2011). Improvements continue to be made, with estimations significantly improving in the coastal
and polar regions (Ray et al., 2019). Poorer results are seen in the coastal region due to poorly resolved bathymetry, the
complexity of ocean tides and due to land contamination of satellite altimetry radar signals (Fok, 2012).

One type of ocean tide model, known as semi-empirical models, is derived from empirical harmonic analysis of satellite
altimetry relative to a reference model. These semi-empirical tide models rely heavily upon satellite altimetry. Recently, signif-
icant advancements have been made to coastal altimetry in several fields including key improvements in correction fields, more
detailed and coastal-specific data editing, and new schemes for radar echo analysis (retracking) (Cipollini et al., 2017). Pic-
cioni et al. (2018) demonstrated an improvement greater than 2 cm for single tidal constituents when using the ALES (Adaptive
Leading Edge Subwaveform: Passaro et al., 2014) retracker that enhances the performances of sea level retrieval in the coastal
region and the corresponding sea state bias correction (Passaro et al., 2018). The continued developments of altimetry in the
coastal region coupled with the increased number of altimetry missions has had a positive impact on the ability of models to
more accurately estimate ocean tides.

EOT11a (Savcenko and Bosch, 2012), the latest in a series of global ocean tide models developed at DGFI-TUM, is an
example of a semi-empirical tide model developed using residual tidal analysis of multi-mission satellite altimetry. EOT11a
exploits altimetry observations of the sea level anomaly (SLA) corrected using a reference ocean tide model (FES2004) to
estimate the tidal harmonic constants. EOT11a showed significant improvements compared to the previous iterations of the
model, EOT08a and EOT10a, with noticeable improvements being seen in the shallow water regions (Savcenko and Bosch,
2012). The model has continued to be developed, with regional studies being conducted by Piccioni et al. (2019b) based on
improvements being made in the coastal region. These improvements are largely driven by the progresses in accuracy and
precision of altimetry in the coastal zone and the use of the updated FES2014 (Lyard et al., 2020) tide model as the reference
model for the residual tidal analysis.

In this paper, the latest global version of the EOT model, EOT20, is presented based on recent developments made in the
field of tide modelling, coastal altimetry and the availability of an increased number of altimetry missions. The objective of
the EOT20 model is to improve the accuracy of tidal estimations in the coastal region while remaining consistent in the open
ocean. In Section 2, a description of the altimetry data used and how EOT20 is produced through residual tidal analysis is
given. Following this, a comparison of the results of the EOT20 model with in-situ observations and other global tide models
is presented in Section 3, with a conclusion and summary given in Section 4.



**Table 1.** The satellite altimeter data used in this study obtained from OpenADB at DGFI-TUM (Schwatke et al., 2014). The corrections listed in Table 2 are applied to all these missions. Most missions are retracked using the ALES retracker (Passaro et al., 2018), marked by [†], with TOPEX and ERS using ocean ranges as provided in SGDR datasets.

| Mission | Cycles | Period |
|---|---|---|
| TOPEX | 001 - 365 | 1992/09/25 - 2002/08/15 |
| TOPEX Extended Mission | 368 - 481 | 2002/09/16 - 2005/10/08 |
| Jason-1 [†] | 001 - 259 | 2002/01/15 - 2009/01/26 |
| Jason-1 Extended Mission [†] | 262 - 374 | 2009/02/10 - 2012/03/03 |
| Jason-2 [†] | 000 - 296 | 2008/07/04 - 2016/07/25 |
| Jason-2 Extended Mission [†] | 305 - 327 | 2016/10/13 - 2017/05/17 |
| Jason-3 [†] | 001 - 071 | 2016/02/12 - 2018/01/21 |
| ERS-1c | 082 - 101 | 1992/03/25 - 1993/12/24 |
| ERS-1g | 144 - 156 | 1995/03/24 - 1996/06/02 |
| ERS-2 | 000 - 085 | 1995/05/14 - 2003/07/02 |
| Envisat [†] | 006 - 094 | 2002/05/14 - 2010/11/26 |

## 2 Residual Tidal Analysis of Satellite Altimetry

The development of EOT20 focused on improving tidal estimations in the coastal region which has been a historically difficult region to accurately estimate tides. EOT20 follows a similar scheme as the former model, EOT11a, consisting of three major steps: the creation of an SLA product including the correction of a reference ocean tide model; the estimation of the residual tides based on this SLA product; and the combination of the reference model with the residual tides to form a new global ocean tide model. These three steps provide a summary of the creation of EOT20 which is expanded in the following sections.

### 2.1 The Altimetry SLA product

The tidal analysis is based on the analysis of SLA derived from satellite altimetry missions (Table 1) obtained from the Open Altimeter Database (OpenADB, https://openadb.dgfi.tum.de, Schwatke et al., 2014). These missions are selected as they provide extended time-series along similar altimetry tracks, with the Jason missions being a follow-on from TOPEX/Poseidon and Envisat a follow-on of the ERS missions, thus providing appropriate data for the estimation of tidal signals. The SLA from these altimetry missions is calculated according to that described in Andersen and Scharroo (2011):

$$SLA = H - R - MSS - h_{geo} \tag{1}$$

where $H$ is the orbital height of the satellite, $R$ the range, $MSS$ the mean sea surface and $h_{geo}$ is the sum of the geophysical corrections (as listed in Table 2). The corrections used are chosen to optimise the estimations of the SLA in the coastal region, without harming the estimations in the open ocean regions.





**Table 2.** List of corrections and parameters used to compute SLA for tidal residuals estimation.

| Parameter | Model | Reference |
|---|---|---|
| ALES sea state bias | ALES | Passaro et al. (2018) |
| ERS sea state bias | REAPER | Brockley et al. (2017) |
| TOPEX sea state bias | TOPEX | Chambers et al. (2003) |
| Inverse barometer before 2017 | DAC-ERA | Carrere et al. (2016) |
| Inverse barometer from 2017 | DAC | Carrère et al. (2011) |
| Wet troposphere | GPD+ | Fernandes and Lázaro (2016) |
| Dry troposphere | VMF3 | Landskron and Böhm (2018) |
| Ionosphere | NIC09 | Scharroo and Smith (2010) |
| Ocean and load tide | FES2014 | Lyard et al. (2020) |
| Solid earth and pole tide | IERS 2010 | Petit and Luzum (2010) |
| Mean sea surface | DTU18MSS | Andersen et al. (2016) |
| Radial error | MMXO17 | Bosch et al. (2014) |

The same corrections are used for each satellite altimetry mission to allow for consistency, with the only differences occurring in the sea state bias correction. The ALES retracker (Passaro et al., 2014) is applied to the Jason missions and the ENVISAT mission based on data availability at the time of running the model, with the other altimetry missions using the REAPER (Brockley et al., 2017) and TOPEX sea state bias corrections (Chambers et al., 2003). This discrepancy in the chosen retracker is designed to benefit from the ability of the ALES retracker in obtaining data closer to the coast which Piccioni et al.

(2019b) showed had positive improvements on the accuracy of the EOT tide model for the major tidal constituents compared to using the other retracked data. Therefore, depending on the retracker that is used, a coastal flag is implemented into the model that limits the distance to the coast. For missions using the REAPER and TOPEX retrackers, a coastal flag is implemented that restricts the use of SLA data up to 7 km from the coastline. For missions using the ALES retracker, however, this distance to the coast is decreased to 3 km (Passaro et al., 2020). An additional flag is also added limiting the absolute value of sea level

anomalies to $\pm$ 2.5 m (Savcenko and Bosch, 2012). The altimetry data is further adjusted to account for radial errors estimated in the cross-calibration of the SLA data using the multi-mission crossover analysis approach presented in Bosch et al. (2014).

As shown in Table 2, the ocean and load tide correction for all missions is the FES2014 oceanic tide model. This is one of the major changes from the previous version of the global EOT model, EOT11a, which used one of the previous versions of the FES model, FES2004. The results of Lyard et al. (2020) showed considerable improvements in FES2014, particularly in the

coastal and shelf regions. These improvements are largely driven by the improved efficiency of data assimilation and accuracy of hydrodynamic solutions. It is, therefore, anticipated that some of the improvements made between the versions of EOT will be due to the improvement in the reference model.

Once all these corrections are applied, the SLA can be estimated for all eleven altimetry datasets which are then gridded onto a triangular grid based on the techniques presented in Piccioni et al. (2019b).





The triangular grids are chosen based on the efficiency of the model and allows for consistency of grid sizes throughout the ocean thus not over-utilising data in regions of dense data availability. For each grid point, SLA values are collected within a variable capsize, with the radius, $\psi$ (in km), of the capsize being a function of latitude ($\varphi$) where $\psi$ = 165 - 1.5 ($\varphi$). Other capsize techniques are available based on a fixed or depth-based capsize but they do not make significant changes to the results of the estimated tidal residuals and depreciate the efficiency of the model. The choice of the variable capsize is also to
compensate for the greater data density available in higher latitudes.

Once collected, the data is then weighted using a Gaussian function based on the distance to the grid point. The use of data from multiple satellite tracks for each node provides a long SLA time series which is important in reducing the aliasing effect and in decorrelating tidal signals with alias periods close to each other (Savcenko and Bosch, 2012). These issues occur due to the low temporal resolution obtained from satellite altimetry (e.g. the Jason missions only sample the same position once every
9.915 days) resulting in tides not being properly estimated. The alias periods for the major tidal constituents for the Jason and the ERS orbits are presented in Smith (1999). The use of nodes with data from multiple altimetry missions, therefore, creates a long enough time series to improve the temporal resolution and reduce possible aliasing effects in the tidal estimations.

## 2.2    Residual Tidal Analysis

From the weighted SLA, residual tidal analysis is performed using weighted least-squares and the Variance Component Es-
timation (VCE) for each grid point of the model. The least-squares approach is applied to the harmonic formula to derive the amplitudes and phases of single tidal constituents from the SLA observations. In EOT20, the seventeen tidal constituents considered and computed are: 2N2, J1, K1, K2, M2, M4, MF, MM, N2, O1, P1, Q1, S1, S2, SA, SSA and T2. The weighted least square analysis follows a standard procedure solving the following equation for each grid point (Piccioni et al., 2019b):

$$\mathbf{x} = (\mathbf{A}^T \mathbf{W} \mathbf{A})^{-1} \mathbf{A}^\tau \mathbf{W} \mathbf{l} \tag{2}$$

with $\mathbf{l}$ being the vector of SLA values, $\mathbf{A}$ the design matrix, $\mathbf{W}$ the diagonal matrix of weights, and $\mathbf{x}$ the vector of unknowns. The unknowns of vector $\mathbf{x}$ are: the in-phase and quadrature coefficients of the tidal constituents being considered; the sea level trend; and the constant values defined as the mean sea level from each specific mission at each node (Piccioni et al., 2019b).

The VCE is implemented to allow for the combination of datasets from multiple satellite missions and allows for appropriate weighting of missions based on their variances to provide a more accurate estimation. The VCE method has been utilised in
a variety of applications and it was introduced into the previous global model, EOT11a (Savcenko and Bosch, 2012), which followed the formulation detailed in Teunissen and Amiri-Simkooei (2008) and Eicker (2008). The VCE is calculated using iterations as the unknowns and the variances, $\sigma$, are initially unknown. The formulation is as follows:

$$\mathbf{N}_x \mathbf{x} = \mathbf{N}_y \tag{3}$$

with $N_x$ and $N_y$ equal to the weighted sum:

$$\mathbf{N}_x = \sum_{m=1}^{k} \frac{1}{\sigma_m^2} \mathbf{N}_{x,m} \qquad\qquad \mathbf{N}_y = \sum_{m=1}^{k} \frac{1}{\sigma_m^2} \mathbf{N}_{y,m}. \tag{4}$$



The variances are iteratively calculated by:

$$\sigma_i^2 = \frac{\Omega_i}{r_i}, \tag{5}$$

where $r_i$ is the partial redundancy with $\Omega_i = \hat{\mathbf{v}}\mathbf{P}_{bb}\hat{\mathbf{v}}$, $\hat{\mathbf{v}}$ being the vector of residuals and $\mathbf{P}_{bb}$ is the dispersions matrix of measurements (Savcenko and Bosch, 2012).

Following the residual analysis, significant residual signals were obtained for all of the tidal constituents. For the M2 and N2 tides (Figure 1), for example, the residual amplitudes can exceed 2 cm with the largest residual tides being seen in the coastal region. Relatively high residual tides are also seen in the western boundary currents, such as the Agulhas Current and the Gulf Stream. The tides observed are the residual elastic tides that consist of both the ocean and the load tides. Therefore, additional analysis has been done to separate these two components for further analysis. There are several techniques that

are described that make this possible (e.g., Francis and Mazzega, 1990) with EOT using the method presented in Cartwright and Ray (1991). This method involves using the complex elastic ocean tide admittance decomposed in complex spherical harmonics as described by Savcenko and Bosch (2012):

$$Z(\phi, \lambda) = \sum_{n,m} a_{n,m} Y_{n,m}(\phi, \lambda). \tag{6}$$

The ocean spherical harmonic admittances of the load tides are described as:

$$Z_l(\phi, \lambda) = \sum_{n,m} \beta_n a_{n,m}^o Y_{n,m}(\phi, \lambda) \tag{7}$$

where $\beta = \frac{\alpha_n}{1+\alpha_n}$ with $\alpha_n = \frac{3}{2n+1} \frac{\rho_w}{\rho_e} h'_n$. The love numbers, $h'_n$, were taken from Farrell (1972) with $\rho_w$ and $\rho_e$ being the density of the ocean and earth. After synthesis of the load tides, the residual ocean tides were computed as the difference between the load and the elastic tide, $Z_o(\phi, \lambda) = Z(\phi, \lambda) - Z_l(\phi, \lambda)$.

## 2.3 Model Formation

Once the ocean and load tide residuals are produced, the full tidal signal is restored by adding the residuals to the FES2014 tidal atlas. The residuals are interpreted onto a 0.125° degree resolution grid with the FES2014 model interpreted onto the same grid resolution. The outputting of the data onto a regular grid is simply done to allow for an easy combination with the FES2014 model as well as to be more user friendly. The north-south extent of the model extends 66°N and 66°S, with the model defaulting to the FES2014 tides in the higher latitudes. This extent is chosen due to the limited altimetry data further

beyond this latitudinal band and the difficulty in modelling the tides in the polar regions. Dedicated studies to the Arctic region such as that of Cancet et al. (2019) demonstrate the complexity of modelling ocean tides in the polar regions and emphasize their importance for satellite altimetry. Future iterations of the EOT model will tackle the estimation of tides in the higher latitudes. A land-sea mask was added to the model based on the GMT tool that uses the GSHHG coastline database (Wessel and Smith, 1996), which is a high-resolution database that contains information about coastlines as well as lake and river

boundaries. These data has a mean point separation of 178 meters which has been interpolated to a 0.125° resolution for use in the EOT20 model.

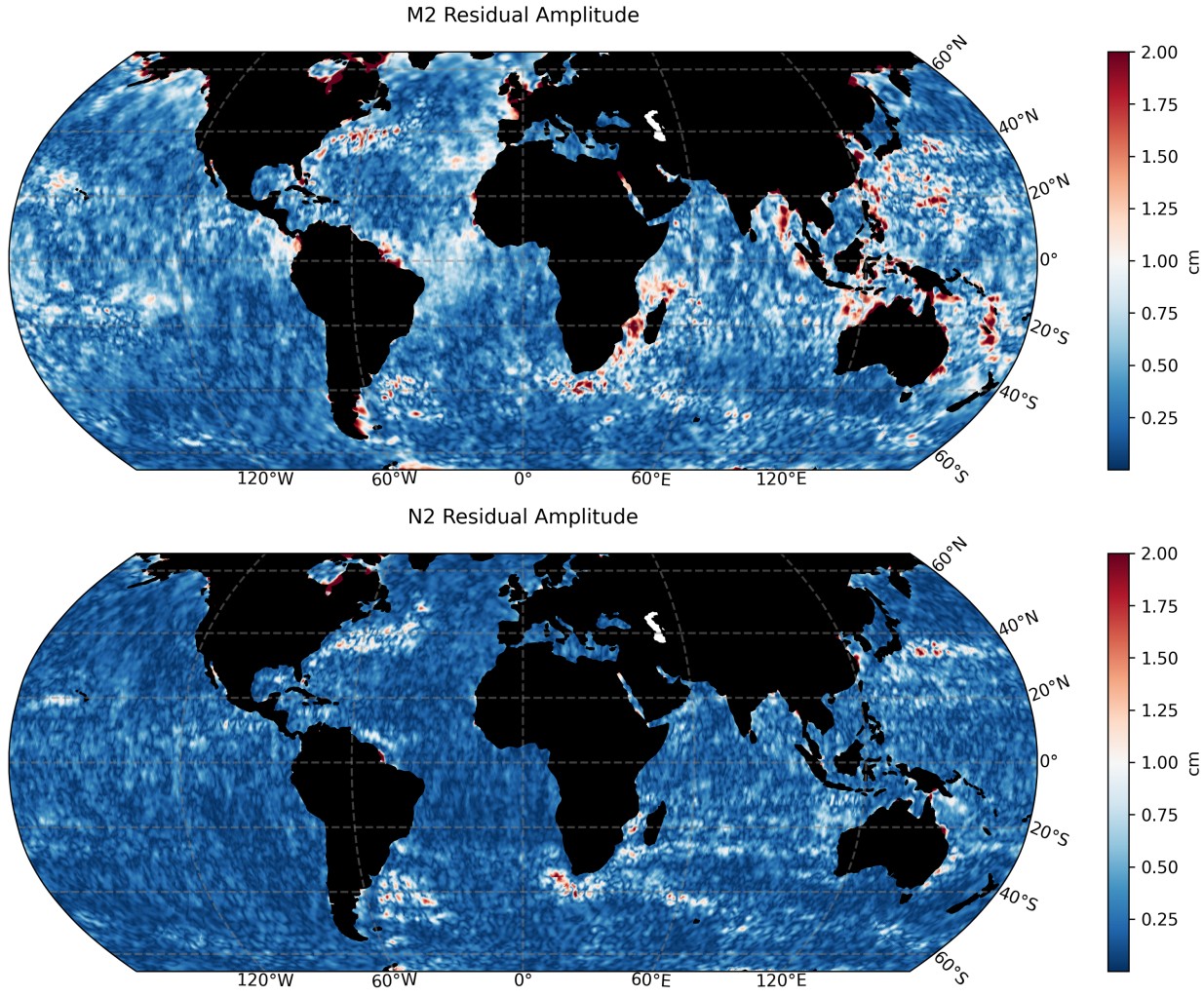

**Figure 1.** Maps of the residual amplitudes of the M2 and N2 tidal constituents as estimated by residual tidal analysis.

In complex coastal regions, such as regions with islands or in semi-enclosed bays, properly defining the coastlines becomes extremely valuable when validating the model against in-situ tide gauges. This is largely a result of artifacts forming when estimating tides in regions where the coastline has not been properly defined. For example, the Cook Strait between the two islands of New Zealand provide a unique coastal structure which shows a sharp change in the amplitude of major tides (e.g. M2, N2, S2 and K2 as shown in Walters et al. (2001)) and, therefore, requires a more accurate coastline definition. Preliminary studies of EOT20 (not shown) demonstrated that for tide gauges within the Cook Strait the root square sum (RSS) difference between the model and tide gauges was reduced by 0.2 cm for the eight major tidal constituents when applying a more accurate land-sea mask. An overall reduction in RSS is seen throughout the ocean when using an accurate land-sea mask.

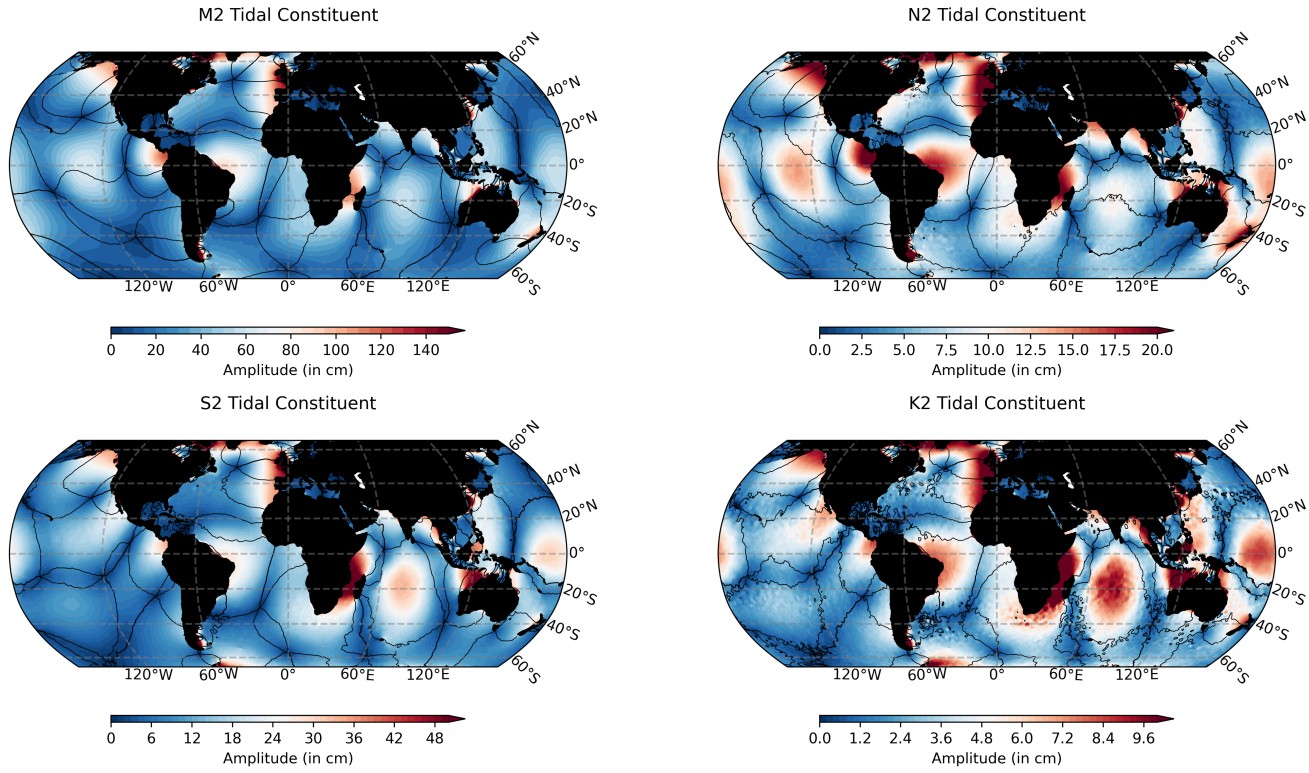

**Figure 2.** The amplitude (in cm) and the phase (in 60° increments) of four ocean tidal constituents produced by the EOT20 model.

## 3 Tide Model Assessment and Validation

### 3.1 The Global EOT20 Model

EOT20 presents global estimations of 17 tidal constituents with these data being available from https://doi.org/10.17882/79489 (Hart-Davis et al., 2021). Global atlas' of both ocean and load tides are provided, containing information about the amplitudes and phases as well as the real and imaginary components for all of the tidal constituents. Here, the ocean (Figure 2 and A1) and load (Figure 3 and A2) tides from EOT20 are presented. Building on from EOT11a an additional four tidal constituents have been estimated in the EOT20 product which are the T2, J1, SA and SSA tidal constituents.

The EOT20 model follows the framework of the EOT11a model when estimating the tide via residual analysis. However, significant changes and additions have been done to EOT20 with the objective of improving coastal estimations. These changes are in the reference tide model used in the residual analysis; the use of more recent developments in coastal altimetry (e.g. the development of the ALES retracker (Passaro et al., 2014)); the increased coverage of satellite altimetry based on the launching of further missions (e.g. Jason-3); the use of an accurate land-sea mask onto the model output data; and using a triangular grid for the residual analysis. These additions all combine to optimise the estimation of ocean tides in the EOT20 model.

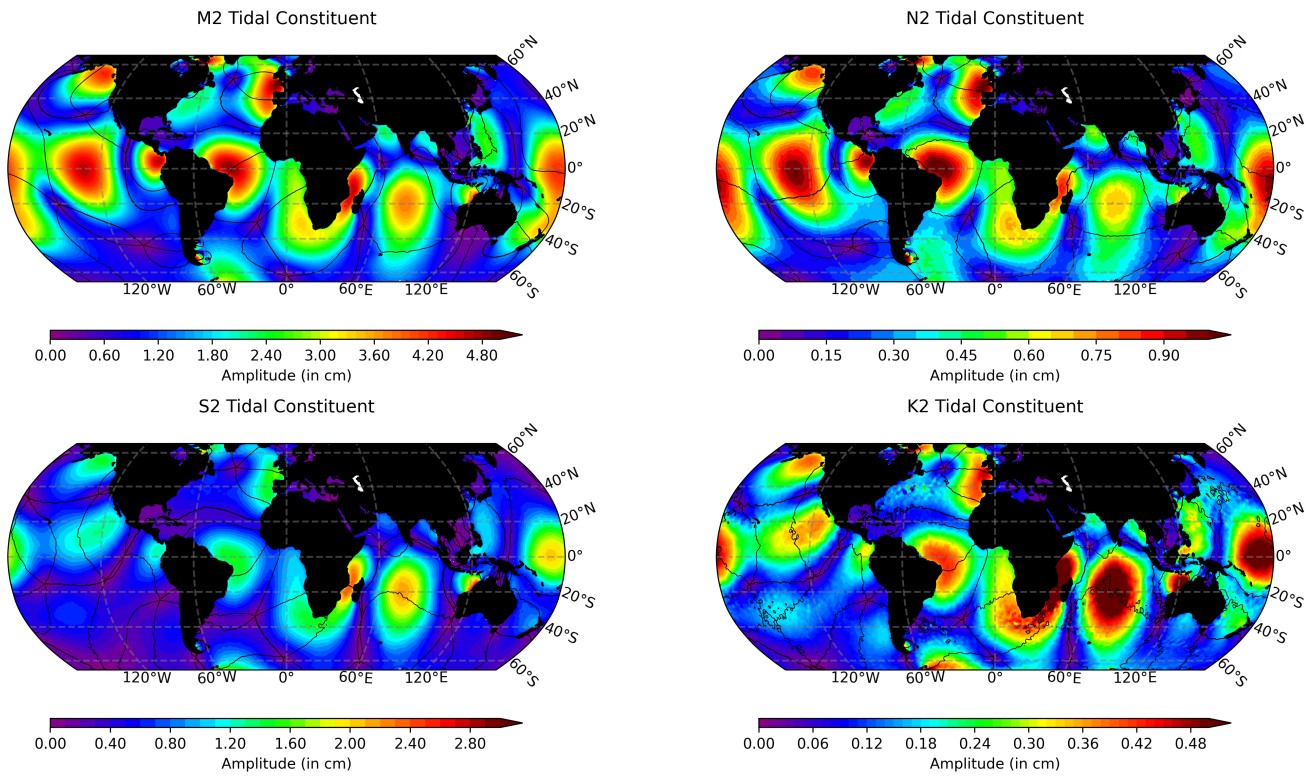

**Figure 3.** The amplitude (in cm) and phase (in 60° increments) of the four load tide constituents produced by the EOT20 model

## 3.2 Tide Gauge Comparison

Since the 1800s, tide gauges have been used to study the ocean tides and the variation in sea level. Over the years, more and

more tide gauges have been installed around the world resulting in a vast array. This comprehensive record of tide gauges can

be used to evaluate the changes in sea level over time as well as better understand the ocean tides. Tide gauges, therefore,

provide a suitable source of data in the validation of ocean tide models, particularly in the coastal region. There are limitations

particularly in the distribution of tide gauges, with certain regions containing a vast number of tide gauges (e.g. in Northern

Europe) and some regions containing little to no data (e.g. the Mozambique Channel). Furthermore, tide gauges are mostly

restricted to the coastal region and, therefore, do not provide sufficient observations of the open ocean region. With that in

mind, Ray (2013) estimated tidal constants from bottom pressure stations in the open ocean regions which has been used to

compare and assess the accuracy of global ocean tide models (Stammer et al., 2014). These data are combined with coastal and

shelf data from Stammer et al. (2014) as well as the TICON dataset (Piccioni et al., 2019a) to create a comprehensive dataset

of tidal constants (shown in Figure 4) to evaluate the accuracy of the EOT20 model throughout the global ocean.

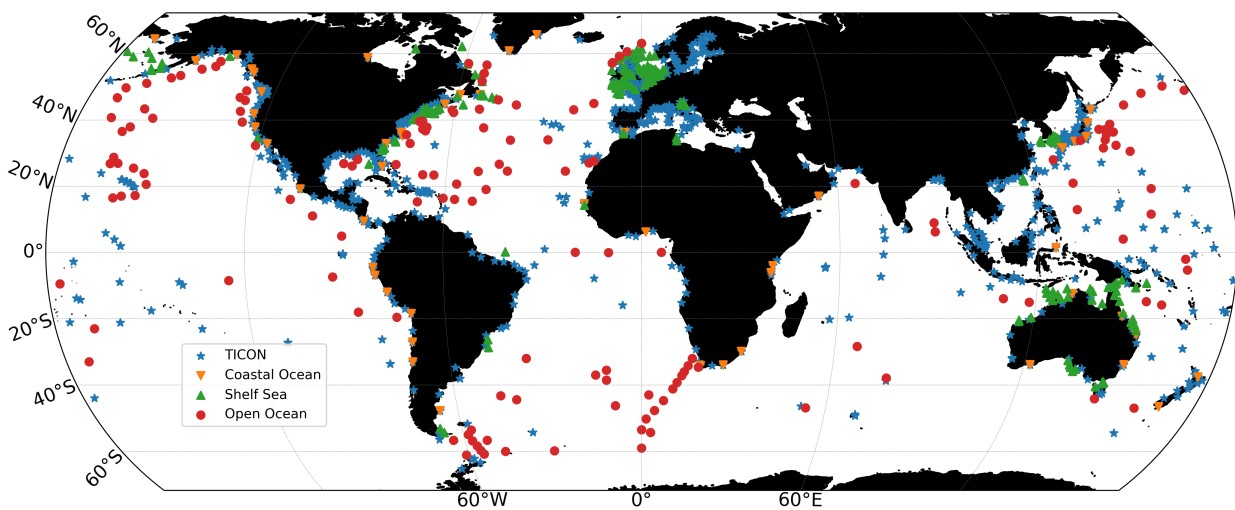

**Figure 4.** The global array of tide gauges and ocean bottom pressure sensors that were used in the validation of the EOT20 model from the TICON dataset and from Stammer et al. (2014): the coastal ocean, shelf sea and open ocean datasets.

As the tide gauge and bottom pressure sensor distributions are already split into coastal, shelf and open ocean tide gauges from Stammer et al. (2014), the TICON dataset is also divided into three regions with the coast being defined as any tide gauges found shallower than 10 m, the shelf defined as being between 10 m to 100 m depth and open ocean being anything deeper than 100 m. This is done to assess how the model performs in the coastal region, a historically difficult region to model accurately. Several major ocean tide models are also compared to the same tide gauges in order to act as reference to the ability

of the EOT20 model. The models used are EOT11a (Savcenko and Bosch, 2012), FES2014 (Lyard et al., 2020), GOT4.8 (Ray, 2013) and DTU16 (Cheng and Andersen, 2017). To provide suitable comparisons, duplicate tide gauges were removed and restrictions were implemented based on the model characteristics (i.e. only tide gauges between 66°S and 66°N were used). This results in 1,226 tide gauges and bottom pressure sensors being available for validation of the models. It should be noted that 230 of the tide gauges used in this study are assimilated into the FES2014 model. The root-mean-square (RMS) and root-

square-sum (RSS) between models and gauges were estimated following the techniques described in Stammer et al. (2014) for the eight major tidal constituents (M2, N2, S2, K2, K1, O1, P1 and Q1) which are commonly available from the tide models used.

       The comparison between EOT11a and EOT20, shows a significant improvement in the EOT20 model for the full dataset (Table 3). This is consistent for all of the tidal constituents, with a major improvement seen in the M2 tide (1.5 cm) and the

S2 tide (0.9 cm). For all of the regions (Figure 5), EOT20 continues to show improvements compared to EOT11a particularly in the coastal region with a mean RSS reduction of 2.5 cm. In the coastal region, EOT20 shows a reduced RMS for all the tidal constants with large reductions occurring again for the M2 (2 cm) and S2 tide (1.3 cm) with significant reductions in the K1 (0.4 cm), K2 (0.3 cm) and N2 (0.7 cm) tidal constituents. Reductions in RSS are seen in the other regions, however, the magnitude is not as large as in the coastal region.





**Table 3.** The RMS, in cm, of the tide gauge analysis of 1,226 tide gauges for the EOT20 model as well as several other global ocean tide models. The values marked in bold indicate the model with the lowest RMS for each row.

| Constituent | GOT4.8 | DTU16 | EOT11a | FES2014 | EOT20 |
|---|---|---|---|---|---|
| M2 | 5.313 | 4.020 | 4.839 | 3.587 | **3.352** |
| N2 | 1.326 | 0.908 | 1.311 | 0.805 | **0.802** |
| S2 | 2.484 | 1.480 | 2.330 | 1.434 | **1.411** |
| K2 | 1.159 | 0.848 | 1.093 | **0.744** | 0.783 |
| K1 | 1.214 | 1.051 | 1.209 | **0.866** | 0.906 |
| O1 | 0.981 | 0.837 | 0.843 | 0.673 | **0.653** |
| P1 | 0.785 | 0.807 | 0.772 | **0.664** | 0.687 |
| Q1 | 0.384 | 0.359 | 0.383 | **0.276** | 0.360 |
| RSS | 6.380 | 4.741 | 5.888 | 4.224 | **4.042** |

This suggests that the adjustments and additions made to the EOT model, such as the incorporation of the ALES retracker in the estimation of the SLA, produce substantial differences to the performance of the model in the coastal region without harming the performances in other regions. EOT20 also shows a reduced RSS when compared to the other global models, particularly compared to the reference model, FES2014. The largest improvement comes in the M2 tidal constituent while the results for the remaining tidal constituents are quite consistent between FES2014 and EOT20. In the coastal region, FES2014

and EOT20 both show significant improvements to the other models, being approximately 1 cm better than the closest model in this region (Figure 5). In the shelf and open ocean regions, all the models generally show similar results to one another with FES2014, EOT20 and DTU16 only varying by a few millimeters. Therefore, the better performance of EOT20 seen in Table 3 can mostly be put down to the results seen in the coastal region.

        This is further highlighted in the TICON dataset which contains significantly more coastal tide gauges compared to the other

two regions. Again, EOT20 shows a substantial reduction in RMS for the M2 tidal constituent of 2.3 mm compared to the next best model (FES2014). For the remaining tidal constituents, EOT20 and FES2014 never vary by more than 1 mm in terms of RMS values. This improvement compared to FES2014 is mainly seen in the coastal region, which is in line with previous regional studies of EOT done using FES2014 as the reference tide model and the ALES retracker (Piccioni et al., 2018).

        In the shelf region, the reduction of RMS in the M2 tide from EOT20 is still seen compared to FES2014 but reduces to 1

mm. The RSS of EOT20 is 0.2 mm higher than that of FES2014 while DTU16 further reduces the RSS by 0.2 mm. This is dataset specific however, with EOT20 performing the best by 0.05 mm in the TICON shelf data and DTU16 performing the best by 0.25 mm in the Stammer et al. (2014) shelf data. This suggests that these three tide models are really on par with one another in the shelf regions. In the open ocean, the similar results continue with the RSS spread between all of the models not exceeding 2 mm. In this region, FES2014 performs the best showing an improved RSS of 1 mm compared to EOT20. For all of

the tidal constituents, EOT20 continues to show similar results to FES2014 with the M2 tidal constituent improvements seen in the other regions of EOT20 (1.42 cm) falling away to be similar to what is seen in FES2014 (1.40 cm).



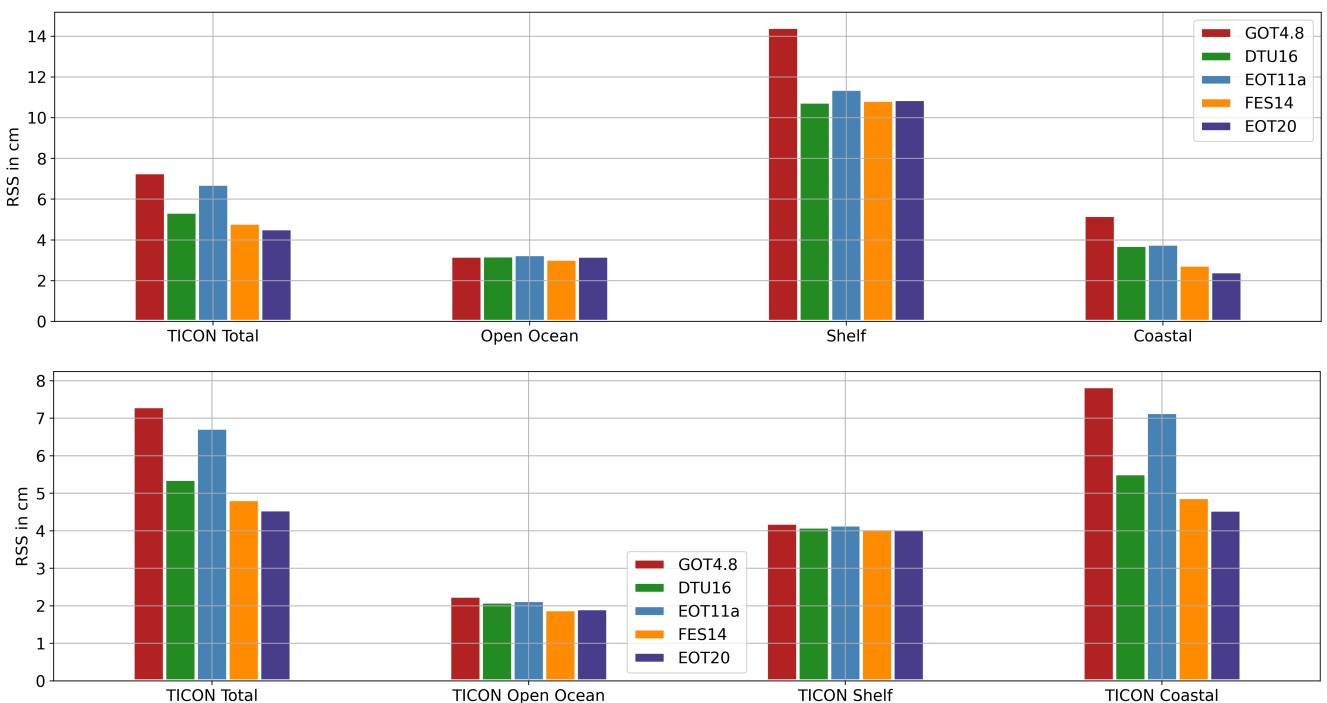

**Figure 5.** (Top) RSS (cm) between the tide gauge databases and the global tidal models, for the eight major tidal constituents. (Bottom) The RSS of subset regions of the TICON database as well as the full database.

The constituents not included in the previous analysis, are compared to the FES2014 model and the TICON tide gauge dataset (presented in Figure A3). Only the TICON tide gauge dataset is used based on the availability of appropriate tidal constituents for the analysis. For eight of the nine tidal constituents, the two tide models show similar results to one another. The SA, solar-annual, tidal constituent shows the largest improvement between FES2014 and EOT20 of 1.9 cm but for both models this is the poorest estimated tide. This constituent is estimated by FES2014 based on free hydrodynamic solutions and does not contain any data assimilation (Lyard et al., 2020). Here, the EOT20 model utilises the extended time-series of altimetry data to make a more accurate estimation of the tide based on the residual analysis, thus providing somewhat of an improvement compared to FES2014. However, it is clear that the solutions of EOT20 are still imperfect due to the poorer performance of the reference tide model and due to the temporal aliasing of this long-period constituent. It should be noted that the assessment of the models using in-situ tide gauges themselves would benefit from additional high-quality extended time-series in order to more accurately estimate long-period constituents. The other free hydrodynamic tidal solutions estimated similarly in FES2014 (MM, MF and SSA) show smaller errors when compared to tide gauges and the differences between the RMS of the two models are significantly reduced.

The S1 tidal constituent is the relatively worst performing tidal constituent from the EOT20 model with an increased RMS of 0.3 cm compared to FES2014. This problematic result is likely influenced by errors from the ionospheric correction, NIC09,

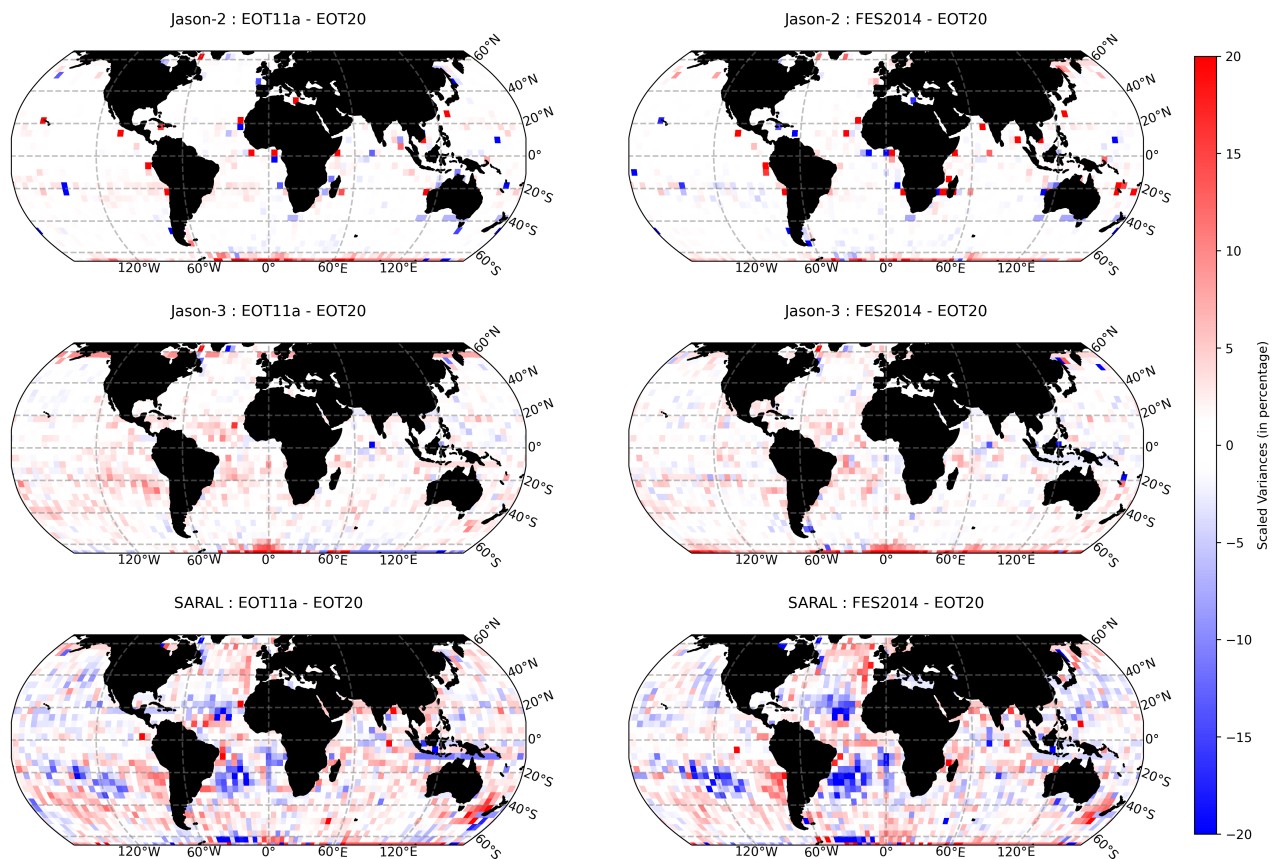

**Figure 6.** The global scaled SLA variances differences for Jason-2, Jason-3 and SARAL in percentages. The colorbar is chosen for ease of understanding with the variance differences scaled to highlight the differences between the results. The colours are chosen so that when there are regions of red colours EOT20 shows a lower variance, while when regions are blue the other tide model (EOT11a or FES2014) has a lower variance.

that is used in the creation of the SLA product which may leak into the estimation of the ocean tides (Ray, 2020). The ionospheric correction used in EOT20 is aimed at optimising the performance of the tide model in the coastal region, however, this may be negatively impacting the estimation of certain tidal constituents, like the S1 tide. Furthermore, Ray and Egbert
(2004) discuss the impact that geophysical corrections (mainly inverse barometer and dry troposphere) have on the estimations of the S1 tide from altimetry data. A future study of the EOT model will investigate the use of different geophysical corrections to optimise the estimation of ocean tides with particular focus on the S1 tidal constituent.

The results of the tide gauge and ocean bottom pressure analysis suggest rather encouraging results from the EOT20 model. The estimated tidal constituents of EOT20 are notably improved compared to the previous EOT11a model. The performance
of the model in the coastal region is noteworthy particularly in the representation of the M2 tidal constituent. Furthermore, the model remains on par with the other global tide models in the open ocean and shelf regions.



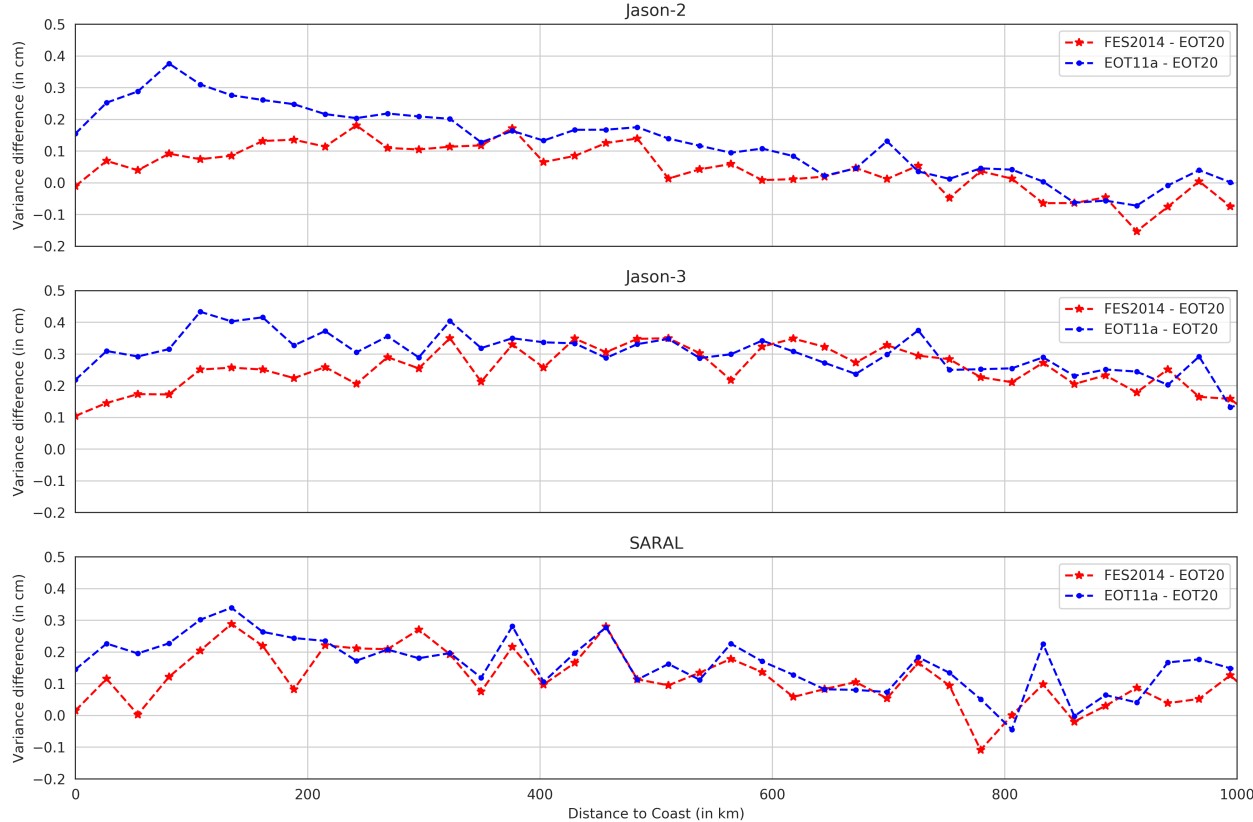

**Figure 7.** A line graph showing the mean SLA variance differences between the tide models as a function of distance to coast (in kilometers) for all three satellite altimetry missions. The red line represents FES2014 - EOT20, while the blue line represents EOT11a - EOT20.

## 3.3 Sea Level Variance Reduction Analysis

In order to further assess the models ability, sea level variance reductions of three satellite altimetry missions were assessed and are presented. As seen in Figure 4, tide gauges and ocean bottom pressure do not provide full coverage of the open ocean

so comparing the sea level variances of ocean tide models provides a suitable assessment of the performances of the models. The missions chosen are Jason-2 and Jason-3 which are used in the residual tide analysis as well as SARAL which is not used in the analysis. A few steps are required in order to estimate sea level variance reduction. First, the along-track SLA is estimated using the corrections listed in Table 2 with the only differences being in the ocean and load tide correction. For this correction, two tide models (EOT11a and FES2014) were used to be compared to EOT20. The SLA for each cycle of all three

missions was then estimated and then gridded onto a four-degree grid. Once done, the variance of each of the SLA products was estimated (Savcenko and Bosch, 2012).





Figure 6, presents the results of the scaled SLA variance differences between the three tide models. For the Jason-2 mission, which is the mission with the most cycles, the SLA variance differences between all tide models are very similar to one another with EOT20 showing an overall mean-variance reduction of 0.54 mm and 0.26 mm when compared to EOT11a and FES2014

respectively. The largest discrepancy is around 60° to 66° south, where EOT20 shows a lower SLA variance compared to EOT11a and FES2014. When looking at how the SLA variance differences change based on the distance to coast for Jason-2 (Figure 7, top), it can be seen that EOT20 shows the largest reduction of variance in the coastal region. This is particularly the case when looking at the differences between EOT11a, with EOT20 reducing the variance by approximately 0.4 cm in the first 100 km from the coast. As they move further from the coast, the difference between the two models begins to reduce and

converge towards zero. The variance difference between FES2014 and EOT20 show similar results. Closer to the coast, EOT20 shows a reduced variance compared to that of FES2014 with differences exceeding 1 mm but as they move further from the coast the difference between the two models converges towards zero. Like with the EOT11a model, FES2014 begins to show a reduced variance compared to EOT20 800 km from the coast.

For the Jason-3 mission, a reduction in SLA variance can be seen from the EOT20 model, with The discrepancies between

the models again being very small (Figure 6). The mean-variance reduction of EOT20 is 0.92 mm and 0.89 mm when compared to EOT11a and FES2014 respectively. The variance reduction can be seen throughout the ocean, with larger reductions in the coastal region (Figure 7, middle). Like in the Jason-2 mission, the variance differences decreases further away from the coast. Although the variance reduction diminishes further from the coast, unlike in the other two missions EOT20 shows continued variance reduction throughout the ocean.

The SARAL mission presents differing results from those seen in the Jason missions. It should be noted that SARAL has considerably less cycles and has a different orbit compared to the Jason missions. However, the results still provide valuable insights into the performances of the models. When looking at the scaled variance differences, the results become a bit more variable between the models with EOT20 showing reductions in variance in regions such as the Indian Ocean and the North Atlantic Ocean but showing increased variance in regions such as the South Atlantic Ocean and the South Pacific Ocean.

Overall, EOT20 shows a mean reduction of variance compared to EOT11a of 1.29 mm despite EOT11a outperforming the model in certain regions. The mean-variance reduction of EOT20 compared to FES2014 is 0.35 mm, however, there are regions where FES2014 shows better performances, particularly in the South Atlantic. Again, the overall reduction in variance is largely driven by the models' performance closer to the coast (Figure 7, bottom) with reductions compared to EOT11a and FES2014 exceeding 3 mm and 2 mm respectively closer to the coast, while these differences reduce towards zero further away

from the coast.




## 4 Conclusions

In this study, an updated version of a global ocean tide model, EOT20, is presented. Model developments were aimed at updating the previous model, EOT11a, with a focus on improving the coastal estimations of ocean tides by utilising recent developments in coastal altimetry, particularly the use of the ALES retracker and sea state bias correction. In the residual analysis, SLA data is gridded into a triangular grid aimed at increasing the efficiency of the model and thus better-describing tides in the coastal and higher latitudinal regions. A further update was in the use of a newer version of the reference model (FES2014) for the residual analysis performed to create the EOT20 model which showed significant improvements to the previous reference model used (Lyard et al., 2020).

To evaluate the performance of the EOT20 model, validation against in-situ observations and through sea level variance analysis was done. First, the models performance was compared with tide gauges and ocean bottom pressure sensors for the eight major tidal constituents. The results suggested that EOT20 showed significant improvements compared to EOT11a throughout the global ocean, with major improvements being seen in the coastal region. Furthermore, when compared to other global ocean tide models, EOT20 showed the lowest overall error for all eight tidal constituents with a major improvement being seen in the M2 tide. This positive performance was largely driven by the improved accuracy of the model compared to observations in the coastal region. In the shelf and open ocean regions, EOT20 was on par with the best tide models in these regions, DTU16 and FES2014. The additional tidal constituents provide valuable data for the creation of the tidal correction used for satellite altimetry. The results of these additions show positive results compared to the FES2014 model but improvements can still be made in determining some of these tides, particularly the S1 tidal constituent. Further investigations will be done at DGFI-TUM into the estimation of additional minor tidal constituents as well as the optimization of the current estimations. The sea level variance analysis continued to show positive results for EOT20. EOT20 reduced the mean variance compared to both FES2014 and EOT11a for all three satellite altimetry missions studied. Again, the largest reason for the improvement was seen in the coastal region with EOT20 showing similar results compared to the other models in the open ocean regions. These results of the new EOT20 model suggest that it will serve as a useful tidal correction for satellite altimetry.

Errors resulting from tide models are considered to be one of the main limiting factors for temporal gravity field determination and the derivation of mass transport processes (Koop and Rummel, 2007; Pail et al., 2016). In the creation of EOT20, a first look into the uncertainties was done but due to the unavailability of uncertainty estimations from the FES2014 model used as the reference model these uncertainties are incomplete and, therefore, are not presented. This is a topic of discussion and future development that will be assessed in future studies.

As the fields of coastal altimetry and ocean tides develop, the ideas and methods of improving the EOT model continue to grow. A clear next step for the EOT model is to assess its ability to estimate tides in higher latitudes by including more satellite missions (e.g. Cryosat-2) and to introduce further data such as synthetic-aperture radar altimetry from Sentinel-3. Furthermore, more recent developments in the estimation of internal tide models (Carrere et al., 2021) suggest that improvements may be made to the estimation of ocean tides from residual analysis when the internal tidal correction is applied to the SLA data. These potential avenues of improvement will be addressed in future iterations of the EOT model.





# 5 Data availability

The ocean and load tides from EOT20 are available at: https://doi.org/10.17882/79489 (Hart-Davis et al., 2021). The GMT data used to create the land-sea mask can be found at: http://gmt.soest.hawaii.edu. The satellite altimetry data used in the model creation can be found at https://openadb.dgfi.tum.de/. The tide gauges from the TICON dataset used in the validation of the tide model, are available at: https://doi.pangaea.de/10.1594/PANGAEA.896587.

*Author contributions.* M.H-D wrote the manuscript and performed the validation of the model. M.H-D and D.D. were involved with designing the study and in interpreting the results. M.H-D and G.P. were responsible for the development of the EOT20 model. C.S. and D.D. were responsible for the appropriate satellite altimetry data and assisted in the variance reduction validation. M.P. is the author of the retracking algorithm and of the sea state bias correction used in the model. F.S. provided the resources making the study possible and coordinates the activities of the research group at DGFI-TUM. All authors read, commented and reviewed the final manuscript.

*Competing interests.* The authors declare that they have no conflict of interest.

*Acknowledgements.* The authors thank NOVELTIS, LEGOS, CLS Space Oceanography Division, CNES and AVISO for providing the FES2014 model.





# Appendix A

**Figure A1.** The amplitude of the remaining ocean tide constituents provided by EOT20.

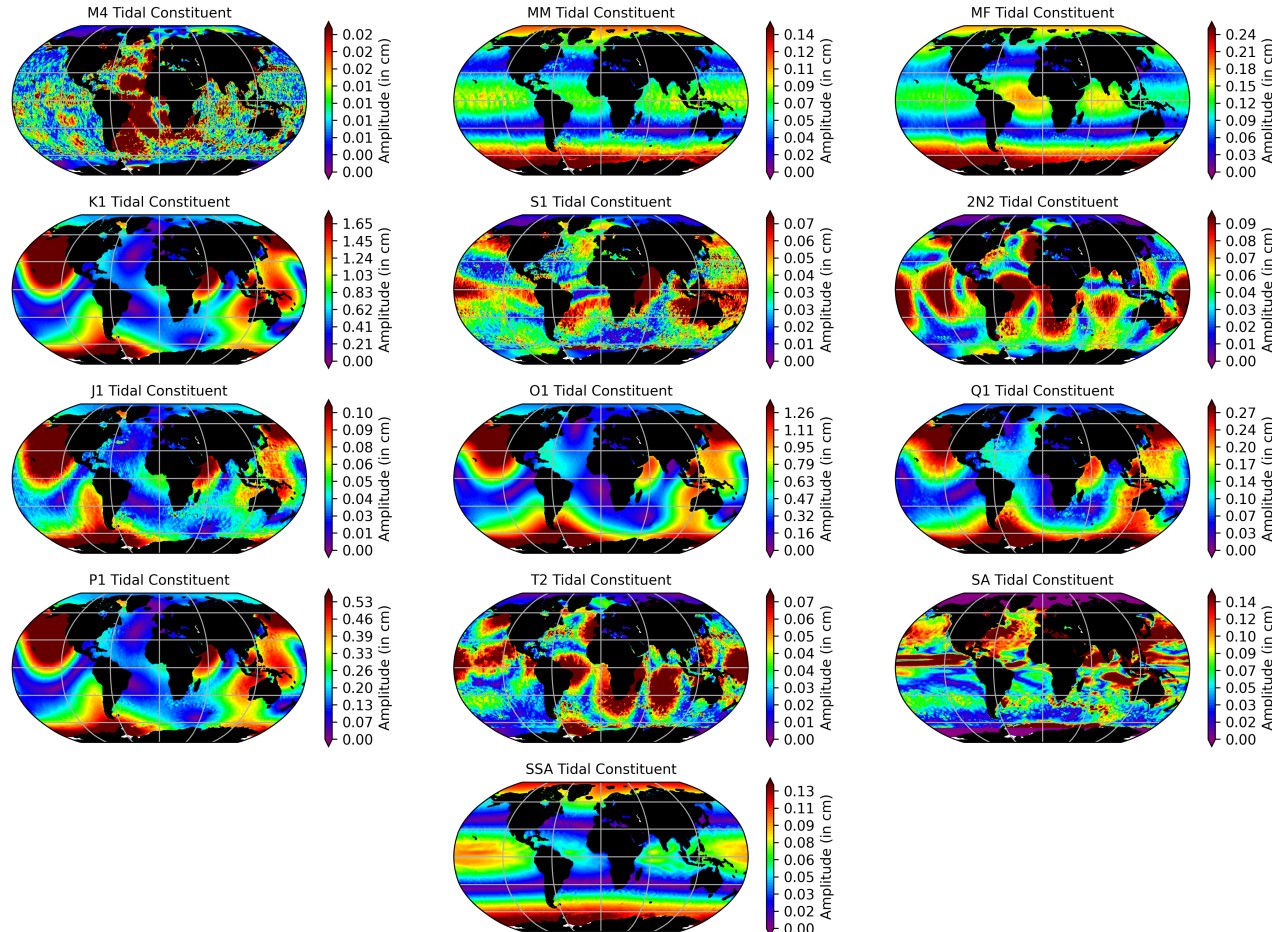

**Figure A2.** The amplitude of the remaining load tide constituents provided by EOT20.



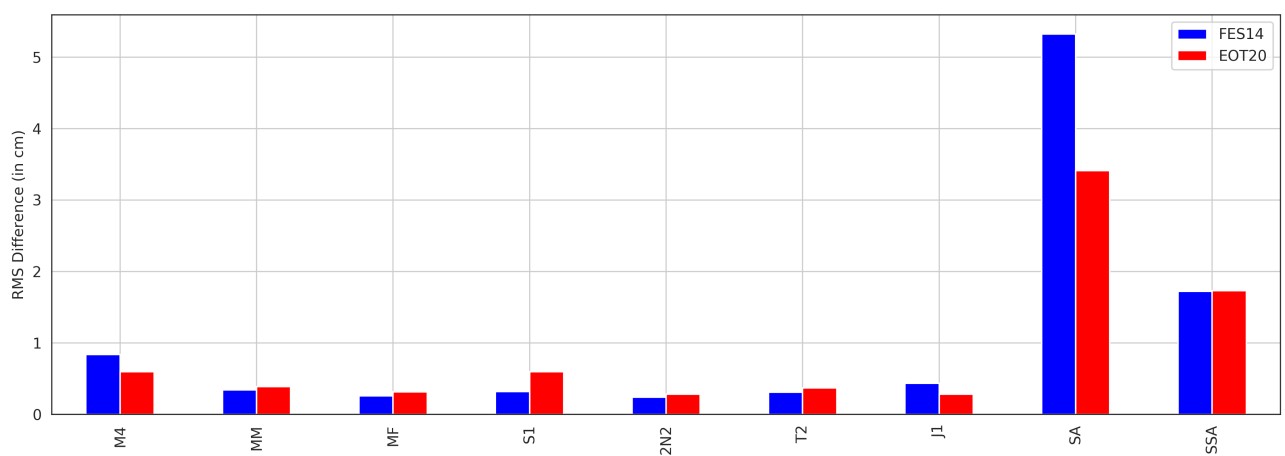

**Figure A3.** The RMS and RSS of the remaining tidal constituents compared to the tide gauge datasets for both FES2014 and EOT20. A total of 1059 of tide gauges were used for this analysis only from the TICON dataset due to the availability of appropriate tidal constituents.





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
