# Peer review of "EOT20: A global ocean tide model from multi-mission satellite altimetry"

_Earth System Science Data, 2021_

## Author Response (AR1)

**Response to Reviewers of Hart-Davis et al 2021:**

We would like to firstly thank both of the anonymous reviewers for the input that they have given and the encouraging comments they have given about the results of the new EOT20 model. The comments and suggestions that have been made have helped make an overall improvement to the manuscript.

Some additional changes have been made to the document, aimed at improving the references and after doing a more recent proof of the English. Additionally, a couple of sentences have been added to line 69 to describe the orbit of the altimetry missions. This is:
*For all of the missions, satellite orbits in ITRF2008 are used. For the ERS and Topex these are taken from GFZ VER11 (Rudenko et al 2016), while for the Jason missions and Envisat CNES GDR-E solutions are used.*

The structure of the comments below are **BOLD** representing the comments of the reviewers, normal text representing our responses, *Italics* representing original text from the manuscript and *Underlined Italics* representing our changes to the manuscript resulting from the reviewers comments.

***Reviewer 1:***

***The paper describes the latest in a series of empirical ocean tide models developed at the DGFI-TUM. It is a clear advance on the earlier EOT11 model, so it is good to have this advancement documented. It is a worthy contribution for this journal. Before it is accepted, however, I recommend certain items be addressed.***

Thank you for these reviews and the recommended items to be addressed. We have answered them below.

**Major item 1. Improvement relative to FES2014.**

***The procedure used is to adopt the FES2014 model as a prior in the altimeter processing and compute tidal estimates relative to this prior, then add back the prior. I think it must be acknowledged that the big improvement from EOT11 to EOT20 may stem from FES2014. The FES2014 is a very good model, which may be hard to improve (especially in non-polar regions, as here). The authors are trying to polish a rock that is already very shiny.***

We agree that the results with FES2014 itself are, in fact, really impressive. With that in mind, we reinforced that a lot of the improvements made in the EOT20 model are caused by progress made by FES2014. In the text we had already stated:

In the introduction at line 50 we state: "*These improvements are largely driven by the progresses in accuracy and precision of altimetry in the coastal zone and the use of the updated*

*FES2014 (Lyard et al., 2020) tide model as the reference model for the residual tidal analysis.*" and in line 91: "*It is, therefore, anticipated that some of the improvements made between the versions of EOT will be due to the improvement in the reference model.*" Then in the conclusion at line 301: "*A further update was in the use of a newer version of the reference model (FES2014) for the residual analysis performed to create the EOT20 model which showed significant improvements to the previous reference model used (Lyard et al., 2020).*" However, with your comments in mind, we have adjusted line 91 to state '*large parts of the improvement*' instead of '*some of the improvements*' to try to further emphasize the importance of FES2014.

**Are they making FES2014 better or worse? According to Table 3, it is half and half. M2 shows improvement; Q1 shows degradation. The other constituents are fairly close. This is by no means a problem. In fact, it is telling us that FES2014 is very good, and it is a challenge to improve, and that is useful to know. I simply think that this should be more clearly acknowledged. (I agree that the altimeter tests in Figure 7 suggest improvement near coasts.)**
**I don't understand how the authors can state (line 308) that "EOT20 showed the lowest overall error for all eight tidal constituents". That statement is contradicted by their Table 3.**

Agreed, generally the differences between EOT20 and FES2014 are very small and, therefore, in some cases in the tide gauge analysis one estimates a certain tide better than another. The M2 tide shows the largest difference between the models, but the differences become relatively larger in the coastal region. Although these differences are not so large in the tide gauge analysis, as you stated this becomes more important in the altimeter tests. With your comments in mind, we have adjusted line 308 to state "*Furthermore, when compared to other global ocean tide models, EOT20 had the lowest overall RSS for the major eight tidal constituents. In particular, improvements are seen in the coastal region, where EOT20 shows a reduced RSS of 0.2 cm compared to the closest model (FES2014). The RMS differences between individual constituents show that EOT20 and FES2014 show clear improvements for all the tides compared to the other global models. EOT20 and FES2014 each had the lowest RMS for half of the major tidal constituents presented, with the largest reduction in RMS being seen in the M2 tide from EOT20.*".

**Major item 2. Handling of the SA and SSA tides.**

**The handling of these two constituents is different from what the altimeter community is accustomed to. The authors solve for the full signals at these periods, which includes both gravitational and meteorological tides (for SA, it is almost entirely meteorological). Usually these kinds of tide models/atlases address only the gravitational tides -- which are equilibrium or very nearly equilibrium. It is fine to include the whole signal, but it must be made clear to altimeter users.**

**For example, I believe the RADS database developers plan to include EOT20 as an optional correction. As it is described here, most altimeter users of this correction will be dismayed to see their seasonal signals in the altimeter data mostly disappear! Again, this is acceptable if users are made aware of it. Many users will not want that.**

**However, because of this, Figure A3 is misleading, as it suggests that EOT20 SA model is much better than FES2014. But by design FES2014 includes only the gravitational tide. By design it leaves out the climate-driven tide. So the comparison is not valid. Moreover, the comparison isn't valid anyway, because the tide gauges include a large IB annual signal that has been removed from EOT20 (and of course would not be part of FES2014). So this comparison figure is not acceptable. These points also apply to SSA, even though the gravity part of it is larger. (SSA also has an IB part.)**

Based on this full comment on the implications of this SSA and SA tides on the analysis of along-track altimetry, we have (1) removed the SSA and SA tides from Figure A3 (2) We have added a sentence into line 171 where we state: *"The SSA and SA tides are included in the EOT20 model data but users should be aware that these tides include the full signal at these periods, i.e. gravitational as well as meteorological tides. Thus, caution should be taken when interpreting the results of the tidal correction when these two tides are included as they will likely remove the seasonal signals seen in the altimeter data."*

*Major item 3.*

**Tide gauge comparisons. It is my understanding of the text that the tide gauge comparisons shown in Figure 5 (top panel) use the same test data as was used by Stammer et al. (2014). However, the results are inconsistent with results reported by Stammer et al. (Models EOT11a and GOT4.8 are common to both papers.) For example, Stammer found that the RSS for EOT11a was about 7 cm (7.17 and 6.87 cm) for the shelf stations, and 6.49 for the coastal stations. This is in disagreement with Figure 5. Either the statistics are being computed differently, or somebody is wrong.**

This is true that the estimation of RMS done in our paper is different to that done in Stammer et al (2014) and the sentence in Line 199 incorrectly refers to this, when it was meant to only refer to the RSS calculation. Stammer et al, estimates the RMS over the cycle length of individual tidal constituents, however, in our paper we look at only the RMS differences between the observations and model data. The techniques we use are described in Piccioni et al (2018) :

$$RMS = \sqrt{\frac{(AM\cos PM - AT\cos PT)2 + (AM\sin PM - AT\sin PT)2}{2}}$$ (please see the latex version in our updated paper)

where A is the amplitude and P is the phase of the  model (M) and tide gauge (T). In order to clarify this and explain the RMS technique further, Line 199 has been adjusted, with an additional sentence added in the end to clarify discrepancies between results:

*"The root-mean-square (RMS) is estimated following Piccioni et al (2018), where*

$$RMS = \sqrt{\frac{(AM \cos PM - AT \cos PT)^2 + (AM \sin PM - AT \sin PT)^2}{2}}$$ *, with A being the amplitude and P the phase of the model (M) and the tide gauge (T). The root-square-sum (RSS) is then estimated for the eight major tidal constituents (M2, N2, S2, K2, K1, O1, P1 and Q1) which are commonly available from the tide models studied, following the technique presented in Stammer et al. (2014). Since Stammer et al 2014 uses a different formulation to estimate the RMS, the RSS values shown here are different to those shown in Stammer et al, however, the relative results of the models compared to tide gauges is the same."*

**Major item 4. Ionosphere model.**

**The authors chose to use the NIC09 model instead of the usual dual-frequency corrections. That is their choice. However, when the models are tested with altimeter data, I think then the NIC09 model should not be used, and instead the usual dual frequency corrections should be used. Use of NIC09 would favor EOT20 (since NIC09 was used in its development), plus almost all altimeter users will be using the usual dual frequency model. (The same might be said of the use of the GPD wet troposphere correction versus one based on onboard radiometer, at least in the open ocean away from coasts.)**

We agree that the ionospheric correction used does influence the resultant tide estimations (as shown in Ray 2020) and stated in line 246 - 247. Although we agree that the correction will likely influence the sea level variance itself, we have designed the study that the variance reductions presented in Figure 6 and 7 are only caused by the ocean tidal correction used. Therefore, the use of a different ionospheric correction will not have an influence on the sea level variance reductions. Furthermore, we want to assess the coastal impact of EOT20 as this is where we expect the biggest discrepancies between EOT11a and FES2014 as shown in the tide gauge analysis (Figure 5) and, therefore, using the NIC09 allows us to get valid SLA data in the coastal region. This allows us to create Figure 7 and analyse the differences between the models a bit more in detail in this region.

**The Smith (1999) reference needs more details. However, there are many review works that tabulate the tidal alias periods; one of them might be a better reference.**

We have removed this reference and included both *(Wang 2004) and Fok (2012).*

*Wang, Y. 2004. Ocean tide modeling in the Southern Ocean. Technical report, Ohio State University. Division of Geodetic Science.*

*Fok, H.S., 2012. Ocean tides modeling using satellite altimetry (Doctoral dissertation, The Ohio State University).*

**The load tide maps should show the whole globe, not just the ocean. Land values are useful for other applications (e.g., GNSS).**

EOT20 only makes estimations of the load tides of the ocean and, therefore, when restoring the full load tide signal the land values are exactly the same as the FES2014 model. With this in mind, we have added a short explanation as to why the load tide maps mask out the land into the figure label of figure 3 and A2. *"It should be noted that EOT20 does not make an estimation for the load tides on land."*

**The VCE explanation is not very clear, partly because several terms are not defined (including even k).**

Thanks for noticing this. We have fixed these inconsistencies by replacing '*i*' in (5) to be '*m*' and adding the following explanation of 'm' and 'k'. In line 119 (with the change made in bold):

*"The VCE is implemented to allow for the combination of datasets from multiple satellite missions and allows for **the** appropriate weighting **of different missions m, m = 1, ..., k, (Savchenko and Bosch, 2012)** based on their variances, to provide a more accurate estimation."*

**Table 2: "inverse barometer" is not really an appropriate description of the DAC. Perhaps "atmospheric loading"? Or even "ocean dealiasing"?**

Table 2 has been appropriately adjusted to state '*Atmospheric loading*'.

**Reviewer 2:**

**The authors present the latest version of the EOT ocean tide model. It is reassuring to see that there are still researchers continuing to improve global ocean tide models. The main result of the new EOT model compared to the previous version is a significant improvement in the coastal regions.**
**The improvements between EOT11a and EOT20 are very impressive. However, this is not the case between EOT20 and FES2014, which was used as the a priori model. It demonstrates the high quality of FES2014. It also shows that the models are converging and future improvements apparently will proceed by small steps.**

**I do support the comments of the other reviewer. In addition, I have two remarks:**

Firstly, thank you for the time taken to present these responses. Please see our response to the first reviewer to address your support of their comments and below we address your questions raised.

1. **Do you calculate the a posteriori covariance of your solutions? What is the final uncertainty? Can we learn something from this information?**

As we state in line 320: "*In the creation of EOT20, a first look into the uncertainties was done but due to the unavailability of uncertainty estimations from the FES2014 model used as the reference model these uncertainties are incomplete and, therefore, are not presented. This is a topic of discussion and future development that will be assessed in future studies.*" We had a first look at the 'residual uncertainty' but, of course, we cannot provide a full uncertainty estimation from the residual analysis. Our first look, showed somewhat expected results such as seeing higher uncertainties in the western boundary currents based on the difficulties to retrieve tidal signals in regions of high mesoscale variability. However, we think improvements need to be made to the uncertainty estimations and an evaluation of whether these residual uncertainties are in fact useful compared to full uncertainty estimations.

2. **Concerning the comparison between the model and the tide gauge observations, what is the uncertainty on those observations? In other words, I am questioning whether differences of 0.2mm or less are significant or not?**

As seen in the paper, the differences are really small, around 0.2 mm, particularly in the shelf and open ocean regions, which is expected from research presented in Stammer et al 2014. The main differences occur in the coastal region, where FES2014 and EOT20 show significant improvements to the other models and EOT20 shows a 0.2 cm reduction compared to FES2014. Speaking on the uncertainties of the tide gauges for TICON, standard deviations of individual constituents are provided and are on average 0.09 mm for the constituents presented. Which considering the differences seen in the TICON analysis (Table 5), these uncertainties do not affect the differences seen between the models. Regarding the Richard Ray dataset, no uncertainties were provided and, therefore, we cannot make a judgement on that in terms of its impact on the analysis. But considering that for the most part the models for the Richard Ray dataset produce results that are close together as most of the tide gauges are found away from the coast and, therefore, there are not much differences between the models, the uncertainties of these measurements should not drastically influence the results. However, because we agree that these information could potentially be valuable to a reader, we have added the following sentence to line 199: "*The TICON dataset provides standard deviations for individual constituents and for the tide gauges used in this study, the average standard deviation is 0.09 mm. No uncertainty estimates are provided in the R. Ray dataset.*"

**Overall, this paper deserves to be published.**

---

## Author Response (AR2)

**(bold: authors comments, non-bold: reviewer comments)**

*To the editor:*

**Through communications with users and internally, we had noted an inconsistency between our version of the model and that maintained on SEANOE. This was the result of a conversion error in the NETCDF file formatting that was not noticed and resulted in a small offset in the downloadable model. It is important to note, that this has been rectified and the issues have no bearing on the results of this manuscript. The updated files as well as the previous error-version of the files are still available on SEANOE.**

*Suggestions for revision*

*Most of the changes made by the authors are good, and I have no problem with most. Let me address several points, based on their replies and changes, in which I have disagreements. These disagreements are not critical, except for the final one, but I want to note them here, because the authors perhaps missed what I was attempting to say.*

**We would firstly like to thank the reviewer for the additional input and we appreciate the effort being done to improve this manuscript as well as helping with ideas for future iterations of the model.**

*Ionosphere. My point was that the use of the NIC09 model in the model testing favored the EOT20 model. The authors disagree, saying "a different ionospheric correction will not have an influence on the sea level variance reductions." I disagree. Let us assume that NIC09 model is inferior to the Jason dual-frequency measurement (likely true). That means the errors in NIC09 were partly absorbed into the EOT20 tide solutions, presumably mostly in S2 wave. Yet when the model tests are run, that error in EOT20 then compensates for the error in NIC09, resulting in a reduced sea-level variance. The FES2014 model would not have absorbed the NIC09 error, so it would result in a larger sea level variance. Thus, the test favors EOT20. Nevertheless, the effect may be small, so I won't insist that the test be redone. (But perhaps the effect is not so small?)*

**We agree with the reviewer on the importance of the ionospheric correction. In order to evaluate the influence of the correction in the sea level variance analysis done in this publication, we did a 'quick' sea level variance analysis for Jason-3 using the dual-frequency measurement correction together with the FES2014 and EOT20 model.**

**The mean variance reduction of EOT20 compared to FES2014 is 0.82 mm when using the Jason dual-frequency measurement (compared to 0.89 mm when using the NIC09 correction), with the mapped differences showing similar results to what was seen in the previous figure for Jason-3 (for example the higher variance reduction of EOT20 around**

**62-66 S). Overall, the variances themselves are smaller using this ionospheric correction, hence showing 'relatively' larger scaled sea level variance differences.**

**So the reviewer is in fact correct that there is an effect of the ionospheric correction visible in SLA reduction. However, the effect is small (0.07 mm) and has no impact on the conclusion of the variance reduction analysis and the assessment of EOT20.**

Load tides. I do not understand the logic of the authors' reply on this point. The reply seems to say that the load tide over land should equal the FES2014 load tide, which cannot be, since any change in the ocean tide results in a change in the load tide (everywhere in principle). But it doesn't really matter, since the load tides in the EOT20 data archive are undefined on land. (It just means that the GNSS community cannot use these published load tide maps.)

**I think there is a misunderstanding here. Of course, we did not question the fact that continental load tides are influenced by ocean tides. Our point was that EOT20 is only containing information over ocean areas (for the ocean as well as for load tides). Thus, we also don't plot this.**

Reference for alias periods. I had noted that the original Smith (1999) reference was incomplete -- e.g., it had no journal name. I also stated that perhaps a review paper would be a better reference, since the alias periods had been reported in earlier studies before Smith (1999). Instead, we now have two obscure departmental reports from Ohio State University! It isn't very important, but I had been thinking of reviews such as C. Le Provost's chapter in the 2001 book "Satellite Altimetry and Earth Sciences". Even Carl Wunsch's recent physical oceanography textbook could be used. But it's the authors' choice.

**We have changed it to be Le Provost's chapter:**

***Le Provost, C.: Chapter 6 Ocean Tides, in: Satellite Altimetry and Earth Sciences, edited by Fu, L.-L. and Cazenave, A., vol. 69 of International Geophysics, pp. 267–303, Academic Press, https://doi.org/https://doi.org/10.1016/S0074-6142(01)80151-0, 2001.***

Tide gauge statistics. This is my one major remaining concern. The tide gauge RMS statistics do not appear correct to me. I'll separate the comment into 3 parts:

(a) Apparently both the EOT20 authors and Stammer et al. (2014) are computing a tidal constituent's RMS difference between model and tide gauge by integrating in time over a tidal cycle. The authors claim they aren't, but their Eq (8) is exactly that. [Note their Reply differs by a factor of sqrt(2)]. (It isn't necessary to cite Piccioni et al. (2018) for Eq (8), as it is a standard result since the integration over the trigonometric functions can be worked out analytically.) But Eq (8) is for a single tide gauge, and perhaps the summation over all gauges is the problem?

The computation done by Stammer et al. isn't clear on this point, because they used a weird notation with an overbar that evidently stands for two different kinds of averaging. Therefore, I

checked with the relevant coauthors of that paper, and I am now fairly certain the Stammer RMS calculation (for a single constituent) was as follows (Latex format, and in the authors' notation):

\begin{equation}
\left[ (2N)^{-1} \sum_{k=1}^N
\left( (A_{M,k}\cos P_{M,k} - A_{T,k}\cos P_{T,k})^2 +
(A_{M,k}\sin P_{M,k} - A_{T,k}\sin P_{T,k})^2 \right) \right]^{1/2}
\end{equation}

for N tide gauges. Is that what is used here? If not, perhaps that would explain why the RMS values here differ. In any event, the new text seems incorrect on these points.

(b) The differences with Stammer et al. are important, and it isn't true (as stated in the revised paper) that the "relative results of the models compared to tide gauges is the same". Referring to Figure 5, we see EOT11a shelf RSS values are inflated by about 50% (about 7 cm in Stammer versus 11 cm here), but GOT4.8 values are inflated by over 100% (about 6.5 cm versus 14 cm). Perhaps there is something else different than the maths. How were the model values interpolated to the tide gauge locations? If this was done inaccurately that could explain some differences. (Interpolation may be more critical for GOT4.8 because the grid interval is larger.)

(c) Figure 5 (top) shows the RSS shelf values are much worse than the coastal values. This also contradicts Stammer et al. In fact, Figure 5 (bottom) seems more realistic in this regard, with coastal stations worse. In the top panel, the EOT20 value for coastal gauges is even lower than the value for the open ocean gauges, which seems very suspect. I think these calculations need to be rechecked.

**We would like to thank the reviewer for this comment as well as the clarifications with the authors of Stammer et al regarding the RMS estimation. In fact, we have now implemented this into our RMS calculation in order to be consistent with Stammer et al 2014 and we can clearly see results similar to what was seen in Stammer et al for the EOT11a and GOT tide models.**

**Furthermore, we reached out to Richard Ray as we were still unable to match the results of the Shelf Sea datasets. The Shelf Sea dataset itself does not contain data for every major tidal constituent at each ocean bottom pressure station, i.e. for some only four major tides are estimated. Ray discussed that the RMS was calculated for individual tides for the observations that were available, with the overall RMS's being based on a different number of tide gauges for each constituent, a problem that does not occur in the open ocean and coastal region. We had only looked at the ocean bottom pressure data that contained data for all 8 major tides. This handling was not discussed in Stammer et al, but thanks to the clarification from Richard Ray we are now following the same approach. However, we only use observations where there is data for at least five constituents, which results in the dropping of five bottom pressure sensors, hence why**

the results of GOT and EOT11 look 'slightly' better in our comparison compared to Stammer et al in the Shelf Seas.

Based on this analysis and the comments of the reviewer, we have changed this result within the text as well as updated Table 3 with the changed RMS results as well as Figure 5 and Appendix A3.

Changes have been made to the text based on changing the numbers and adding a few more explanations, from lines 207 - 254.